# Putnam-AXIOM: A Functional and Static Benchmark for Measuring Higher Level Mathematical Reasoning

**Aryan Gulati**
Department of Computer Science
Stanford University
aryangul@stanford.edu

**Brando Miranda**
Department of Computer Science
Stanford University
brando9@stanford.edu

**Eric Chen**
Department of Mathematics
Stanford University
ericc97@stanford.edu

**Emily Xia**
Department of Mathematics
Stanford University
emxia18@stanford.edu

**Kai Fronsdal**
Department of Computer Science
Stanford University
kaif@stanford.edu

**Bruno de Moraes Dumont**
Department of Mathematics
Stanford University
bdumont@stanford.edu

**Sanmi Koyejo**
Department of Computer Science
Stanford University
sanmi@stanford.edu

## Abstract

As large language models (LLMs) continue to advance, many existing benchmarks designed to evaluate their reasoning capabilities are becoming saturated. Therefore, we present the Putnam-AXIOM Original benchmark consisting of 236 mathematical problems from the William Lowell Putnam Mathematical Competition, along with detailed step-by-step solutions. To preserve the Putnam-AXIOM benchmark's validity and mitigate potential data contamination, we created the Putnam-AXIOM Variation benchmark with functional variations of 52 problems. By programmatically altering problem elements like variables and constants, we can generate unlimited novel, equally challenging problems not found online. We see that almost all models have significantly lower accuracy in the variations than the original problems. Our results reveal that OpenAI's o1-preview, the best performing model, achieves merely 41.95% accuracy on the Putnam-AXIOM Original but experiences around a 30% reduction in accuracy on the variations' dataset when compared to corresponding original problems. The data and the evaluation code are available at https://anonymous.4open.science/r/putnam-axiom-B57C/.

## 1 Introduction

The ability for Large Language Models (LLMs) to reason about complex problems has a plethora of applications in many fields such as economics [Zhang et al., 2024], drug discovery [Bran et al., 2023], and even simulations of human behavior and society [Park et al., 2023]. The prominence of this ability has led to significant development in the performance of LLMs on many reasoning benchmarks.

**Outpacing Current Evaluations.** Indeed, advanced models like GPT-4 [OpenAI, 2023] and Gemini Ultra [Team, 2023] have even surpassed human-level performance on many benchmarks like MMLU [Hendrycks et al., 2020] and MMMU [Yue et al., 2023]. Similarly, LLMs have seen astonishing

Submitted to 38th Conference on Neural Information Processing Systems (NeurIPS 2024) Workshop on MATH-AI.

progress in other challenging benchmarks like GSM8K [Chen et al., 2022] and MATH [Hendrycks et al., 2021], with SOTA models attaining nearly $90\%$ accuracy on MATH [Lei, 2024] and nearly perfect accuracy on GSM8K [Zhong et al., 2024]. Though this progress is a testament to the rapidly evolving ability and utility of LLMs, it also presents a large problem: Existing datasets are no longer sufficient to evaluate the reasoning abilities of LLMs.

**Data Contamination.** Compounding this issue is one of the most significant problems facing evaluation datasets today, i.e., data contamination. As LLMs are increasingly trained on more of the internet, an increasing number of the open-source problems used in evaluation benchmarks are incorporated in the training data of these models. A model can therefore display artificially high "reasoning ability" by simply memorizing the answers it has seen undermining evaluation integrity.

To address these limitations, we introduce the Putnam-AXIOM (**A**dvanced e**X**amination of **I**ntelligence in **O**perational **M**athematics) dataset, a novel and challenging compilation of high-level mathematics problems sourced from the prestigious William Lowell Putnam Mathematical Competition, an annual mathematics competition for undergraduate college students in North America which requires advanced mathematical reasoning and covers a wide range of university-level mathematical concepts. Further, we also introduce functional variations of this AXIOM dataset to combat data contamination taking inspiration from the solution employed by Srivastava et al. [2024]. These are small variations of questions on the Putnam that are equally difficult as the Putnam but unavailable anywhere on the internet. AXIOM enables fully automated evaluations by requiring models to provide final answers within "\boxed{}" brackets which can then be extracted and compared to the ground truth final solution using an equivalence function[1]. This approach eliminates the need for human evaluation, allows for complex open-ended answers, and avoids the limitations of multiple-choice formats, thus maintaining rigor while enabling scalability.

Initial evaluations on Putnam-AXIOM demonstrate its difficulty with OpenAI o1-preview scoring less than half at $41.95\%$, while GPT-4o achieves only $17.80\%$. Even math-specialized models such as Qwen2-Math-7B and Qwen2-Math-7B-Instruct perform poorly, scoring $5.51\%$ and $11.8\%$ respectively. Performance further declines on functional variations of Putnam-AXIOM, which include significant drops for most models, decreasing by 20-30% in relative performance. These low scores underscore Putnam-AXIOM's utility for measuring LLMs' advanced reasoning capabilities, while the variations scrutinize true reasoning skills by exposing the models' reliance on memorization.

## 2 Methods

### 2.1 Putnam-AXIOM Original Dataset

**Dataset.** The Putnam-AXIOM Original Dataset contains 236 problems curated from the William Lowell Putnam Mathematical Competition posed between 1985 and 2023. These problems were selected based on their ability to yield final "\boxed{}" solutions ensuring compatibility with our automated evaluation. The dataset encompasses various subjects within university-level mathematics categorized into 11 distinct domains - Geometry, Algebra, Trigonometry, Calculus, Linear algebra, Combinatorics, Probability, Number theory, Complex numbers, Differential equations and Analysis.

To maintain a consistent and rigorous evaluation, each problem retains its original exam ID, which indicates its difficulty level (A or B for sitting, 1-6 for increasing complexity). This categorization helps in evaluating subject-specific understanding and overall problem-solving skills at different levels of complexity. The dataset is formatted using LaTeX to accurately capture the complex equations and symbols the problems employ. Additionally, we utilize Asymptote vector graphics for encoding mathematical figures and diagrams to ensure language models can process visual elements directly. Further, we standardized the placement of boxed answers by relocating them to the end of each solution string to minimize unintended emergent behaviors leading to evaluations that are less "harsh" or prone to penalizing the model for formatting deviations rather than actual comprehension.

**Model Assessment.** Drawing inspiration from the MATH dataset by [Hendrycks et al., 2021], which demonstrated the effectiveness of using boxed answers for evaluating mathematical understanding in LLMs, we similarly create a dataset with final solutions being wrapped in \boxed{} commands. Boxed answers allow for an exact match criterion rather than relying on approximate heuristics by

---

[1]For instance, the equivalence function would evaluate the answers 0.5, 1/2, and \frac{1}{2} as equal

simply parsing the LLM generated string solution for the value within the box, thereby enhancing reliability and consistency of the evaluation process while being quick and cost-effective. To further ensure fair evaluation, we implemented an equivalence function that homogenizes similar answers, addressing both simple string inconsistencies and complex mathematical equivalences like $(x + 1)^2$ and $x^2 + 2x + 1$ or numerical expressions such as \frac{1}{2}, 1/2, and 0.5 and equating them.

**Modified Boxing.** Given the complex nature of certain Putnam questions, some problems do not lend themselves to simple, singular boxed answers. Instead, they often include conditions, multiple possible answers, varied answer formats and elaborate proofs. These original questions would have necessitated costly and difficult human evaluations which we seek to avoid. To address this, we modified these questions by adding a trivial next step to the original questions, changing the solution accordingly. This additional step was designed so as to ensure that solvers reached the same conclusions and insights necessary to solve the problem, but then needed to perform a simpler computation to get a simplified, boxable answer. We provide an example of such a change in Figure 3. By incorporating this minor modification, we preserved the inherent difficulty and complexity of the original problems while making the answers suitable for our boxed answer evaluation criteria.

## 2.2 Putnam-AXIOM Variation Dataset

Models trained on snapshots of the internet have likely encountered Putnam questions, potentially inflating their performance on the Putnam-AXIOM Original dataset. Therefore, drawing inspiration from Srivastava et al. [2024], we introduce functional variations of select problems from Putnam-AXIOM Original providing an effective way of evaluating models that have been trained on the entire internet by taking advantage of weaknesses in model memorization. These variations are classified into two types.

1. **Variable Change.** The simplest variation is a variable change, where variable names are altered and the final answer is unvaried. Variable changes slightly modify the problem from its original statement, which models could have trained on.

2. **Constant Change.** Constant changes modify numeric properties of the question, altering constants within the step-by-step solution and the final answer. Constant changes significantly transform the problem from its original statement, challenging models to perform complex reasoning on how the changes affect the solution and final answer, as in the example from Figure 4.

**Variational Dataset Description.** We created functional variations for 52 Putnam-AXIOM questions, considering limitations such as problem-specific constants, non-generalizable solutions, and questions lacking constants or boxable answers. The dataset includes 26 constant+variable and 26 variable-only changes. We rephrased problem statements while maintaining the core task to prevent pattern recognition by LLMs. Each variation can generate infinite unique, equally difficult snapshots, offering a sustainable evaluation method. To evaluate various SOTA models, evaluators are expected to generate snapshots (instances of the infinite potential variations) of the variation dataset by running the generation code.

## 2.3 Model Evaluations

Using the LM Harness Evaluation framework [Gao et al., 2024], we evaluated several open-source and proprietary SOTA LLMs. Models were prompted to provide answers in \boxed format, which were then compared to Putnam ground truths with an exact final answer match. We evaluated the 236-question Putnam-AXIOM Original dataset once. For the variation dataset, we conducted five trials, each using a randomly selected variation snapshot and its corresponding 52 original questions. We then calculated mean accuracy and 95% confidence intervals.

# 3 Results and Analysis

## 3.1 Putnam-AXIOM Model Performance

Table 1 presents Putnam-AXIOM Original dataset accuracies. Most models score below 10%, with even NuminaMath, the AI Mathematics Olympiad winner [Investments, 2024], achieving only 4.66%.

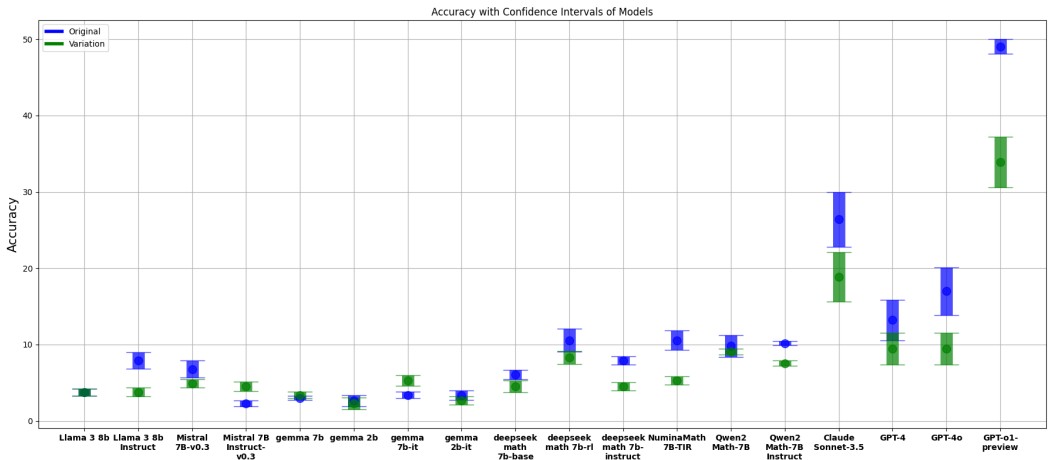

Figure 1: **Drop in accuracies on Putnam-AXIOM Variation vs Original questions is statistically significant** for nearly all models. Figure shows mean accuracies with $95\%$ confidence intervals.

These low accuracies underscore AXIOM's difficulty. Figure 1 contrasts Putnam-AXIOM Variation dataset mean accuracies with the $52$ corresponding original questions, along with the confidence intervals across the five variation snapshots with the average accuracies in Table 2. Original accuracies typically surpass variation accuracies. For models like o1-preview, GPT-4o, Claude-3.5 Sonnet and NuminaMath-7B-TIR, non-overlapping confidence intervals reveal statistically significant differences, indicating artificially inflated performance on original questions due to data contamination. Looking at the numbers highlights significant accuracy declines across models: GPT-4o shows the steepest drop at **44%**, followed by o1-preview at **30%**, GPT-4 at **29%**, and Claude-3.5 Sonnet at **28.5%**.

| Model | Original (Final Accuracy) | |
| :---: | :---: | :---: |
| | Score | Percentage (%) |
| Gemma-2B-Base | 7/236 | 2.97 |
| Gemma-7B-Base | 9/236 | 3.81 |
| DeepSeek-Math-7B-Base | 14/236 | 5.93 |
| Qwen2-Math-7B-Base | 13/236 | 5.51 |
| NuminaMath-7B-Base | 11/236 | 4.66 |
| Mistral-7B-v0.3-Base | 7/236 | 2.97 |
| Llama-3-8B-Base | 9/236 | 3.81 |
| Gemma-2B-Instruct | 2/236 | 0.85 |
| Gemma-7B-Instruct | 8/236 | 3.38 |
| Qwen2-Math-7B-Instruct | 28/236 | 11.86 |
| DeepSeek-Math-7B-Instruct | 12/236 | 5.08 |
| Mistral-7B-Instruct-v0.3 | 8/236 | 3.38 |
| Llama-3-8b Instruct | 10/236 | 4.23 |
| DeepSeek-Math-7B-RL | 19/236 | 8.05 |
| Claude-3.5 Sonnet | 38/236 | 15.96 |
| GPT-4 | 22/236 | 9.32 |
| GPT-4o | 42/236 | 17.80 |
| o1-preview | 99/236 | 41.94 |

Table 1: **Putnam-AXIOM Original Results**.

## 3.2 LLM Error Analysis

Though we used automated evaluations for efficiency, a manual review of model responses on Putnam-AXIOM Original provides deeper insights into models' reasoning and errors. We selected the two best-performing models, GPT-4o and OpenAI o1-preview, as they likely exhibit the strongest reasoning abilities. Our goal is to analyze this reasoning in greater depth.

**OpenAI o1-preview Performance:** Out of all models, we see that OpenAI o1-preview performed the best on Putnam-AXIOM Original, receiving $41.9\%$ boxed accuracy ($99/236$) while other models received less than $20\%$. Analyzing the answers, we see that most of the OpenAI o1-preview responses followed generally the same logical path as the ground truth solution. However, several of these questions contained logical mistakes and inconsistencies. The biggest discrepancy between model responses and the ground-truth solution was a general lack of mathematical rigor. Whereas the ground truth solution will make claims to advance its solution then prove those claims step-by-step, o1-preview will often make and use claims without justification. While this does succeed in getting to the correct boxed final answer, these unjustified claims would receive little credit when marked by a human grader. A large part of the difficulty of mathematical reasoning is being logically airtight throughout the entire solution; thus, though o1-preview shows promise, there are still evident flaws in its mathematical reasoning abilities. In several solutions like Figure 7, for instance, o1-preview correctly identified the maximal or minimal value of a variable, but failed to provide sufficient proof that the value it provided was indeed the maximum or minimum.

**GPT-4o Performance:** Like the o1-preview, GPT-4o mostly followed correct logical reasoning for most of its solutions. For GPT-4o, the biggest discrepancy between model responses and the ground-truth solution is the same general lack of mathematical rigor throughout most of the solutions. An example of this lack of rigor is shown in Figure 8, where GPT-4o makes the claim that a rectangle gives the minimal area subject to a set of constraints without any justification. In addition to issues with rigor, GPT-4o also displayed logical leaps and incoherent reasoning, as displayed in Figure 9 where the model simply assumes that an answer is correct. These logical leaps are symptomatic of an issue in the GPT-4o's CoT reasoning, as the model prioritizes reaching the final answer rather providing a rigorous logical output.

**General Analysis:** Beyond GPT-4o and the o1-preview, we wanted a general overview of the reasoning behaviors of models. To do so, we chose the best-performing open-source models, DeepSeek-Math-7B-RL, Qwen2-Math-7B, and NuminaMath-7B. We tend to see that open-source models are much more error-prone than the proprietary models we evaluated earlier. In general, we notice that open-source models are subject to the same lack of mathematical rigor. However, this rigor issue is overshadowed by major calculation errors, hallucinated/irrelevant information, misunderstandings of the problem, and logical jumps. For instance, in Figure 10, NuminaMath simultaneously makes a calculation, irrelevancy, and misunderstanding error when writing the last step of its solution; in Figure 11, the model makes false assumptions about functions defined in the problem; in Figure 12, the model completely removes a crucial part of the problem and proceeds to an incorrect final solution.

# 4 Conclusion

In this paper, we present Putnam-AXIOM, a novel challenging benchmark of 236 problems from the Putnam examination for evaluating reasoning capabilities of large language models. Our dataset allows for automated evaluations with an equivalence function. While SOTA LLMs already have saturated performance on benchmarks like MATH, they still struggle with successfully answering questions in Putnam-AXIOM. To address potential data contamination issues, we introduce Putnam-AXIOM Variations, altering the variable names, constant values, or the phrasing of the question to create a potentially infinite number of problems not found anywhere on the internet. We notice that for most problems, models get significantly worse on the variations than they do the corresponding original questions. Our dataset fills the void opened by rapid progress in model reasoning capabilities. We hope that our benchmark will accelerate future research into artificial reasoning.

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

## A  Appendix / supplemental material

### A.1  Legal Compliance

We collect and modify various problems from the William Lowell Putnam Competition to create the original and variation datasets of Putnam-AXIOM. Putnam problems are created by the Mathematical Association of America (MAA), which is also the source of the AMC and AIME problems used in the MATH dataset [Hendrycks et al., 2021]. Like Hendrycks et al. [2021], we do not in any form seek to monetize or commercialize Putnam problems—only to utilize them for academic purposes.

Our use of the Putnam problems to create an evaluation dataset completely falls under the "research" section of Fair Use. Indeed, according to Section 107, of the U.S. Copyright Act [USC, 1976], our work certainly qualifies as Fair Use for the following reasons:

1. Our use of MAA problems is *only* for academic research purposes. We do not monetize or commercialize the problems.

2. Our use of Putnam problems as a reasoning evaluation benchmark for large language models is significantly different from their original use as competition problems.

3. Our use of Putnam problems is transformative. As detailed in Section 2 above, we have transformed the questions to be answered with a single numerical or algebraic "boxed answer" We have altered all of the solutions so that the final boxed answer lies at the end of the solution (so as to encourage models to explain their rationale before outputting a solution). We have also standardized the solutions: If there are many solutions given, we only use the first; if there are any references irrelevant to mathematics necessary to understand and solve the problem (such as comments like "Communicated by ..."), we have removed those.

4. Our use of Putnam problems to construct a benchmark has no effect on the demand for or supply of Putnam problems in the William Lowell Putnam Competition. The existence of our dataset does not alter the value of the original problems—as those are already freely available online—nor does it influence the market of future competitors/problem writers.

**Problem:** Let $F_m$ be the $m$th Fibonacci number, defined by $F_1 = F_2 = 1$ and $F_m = F_{m-1} + F_{m-2}$ for all $m \geq 3$. Let $p(x)$ be the polynomial of degree 1008 such that $p(2n + 1) = F_{2n+1}$ for $n = 0, 1, 2, \ldots, 1008$. Find integers $j$ and $k$ such that $p(2019) = F_j - F_k$ and give the answer in the form $j/k$.

**Solution:** More generally, let $p(x)$ be the polynomial of degree $N$ such that $p(2n + 1) = F_{2n+1}$ for $0 \leq n \leq N$. We will show that $p(2N + 3) = F_{2N+3} - F_{N+2}$.
Define a sequence of polynomials $p_0(x), \ldots, p_N(x)$ by $p_0(x) = p(x)$ and $p_k(x) = p_{k-1}(x) - p_{k-1}(x + 2)$ for $k \geq 1$. Then by induction on $k$, it is the case that $p_k(2n + 1) = F_{2n+1+k}$ for $0 \leq n \leq N - k$, and also that $p_k$ has degree (at most) $N - k$ for $k \geq 1$. Thus $p_N(x) = F_{N+1}$ since $p_N(1) = F_{N+1}$ and $p_N$ is constant.
We now claim that for $0 \leq k \leq N$, $p_{N-k}(2k + 3) = \sum_{j=0}^{k} F_{N+1+j}$. We prove this again by induction on $k$: for the induction step, we have

$$p_{N-k}(2k + 3) = p_{N-k}(2k + 1) + p_{N-k+1}(2k + 1)$$

$$= F_{N+1+k} + \sum_{j=0}^{k-1} F_{N+1+j}.$$

Thus we have

$$p(2N + 3) = p_0(2N + 3) = \sum_{j=0}^{N} F_{N+1+j}.$$

Now one final induction shows that $\sum_{j=1}^{m} F_j = F_{m+2} - 1$, and so $p(2N + 3) = F_{2N+3} - F_{N+2}$, as claimed. In the case $N = 1008$, we thus have $p(2019) = F_{2019} - F_{1010}$. We thus prove that $(j, k) = (2019, 1010)$ is a valid solution with the final answer thus being $\boxed{2019/1010}$.

| **Year:** 2017 | **ID:** A6 | **Final Answer:** 2019/1010 |

Figure 2: An example problem in Putnam-AXIOM. Solving this problem requires non-trivial constructions and multiple advanced reasoning chains. The format of the final answer is specified in the problem statement to make comparison simpler.

**Problem:** Determine which positive integers $n$ have the following property: For all integers $m$ that are relatively prime to $n$, there exists a permutation $\pi\colon \{1, 2, \ldots, n\} \to \{1, 2, \ldots, n\}$ such that $\pi(\pi(k)) \equiv mk$ (mod $n$) for all $k \in \{1, 2, \ldots, n\}$.

**Problem:** Determine the sum of the first $k$ positive integers $n$ (in terms of $k$) which have the following property: For all integers $m$ that are relatively prime to $n$, there exists a permutation $\pi\colon \{1, 2, \ldots, n\} \to \{1, 2, \ldots, n\}$ such that $\pi(\pi(k)) \equiv mk$ (mod $n$) for all $k \in \{1, 2, \ldots, n\}$.

**Solution:** The desired property holds if and only if $\boxed{n = 1 \text{ or } n \equiv 2 \pmod 4}$. Let $\sigma_{n,m}$ be the permutation of $\mathbb{Z}/n\mathbb{Z}$ induced by multiplication by $m$; the original problem asks for which $n$ does $\sigma_{n,m}$ always have a square root.

$\cdots$

By Lemma 1, $\sigma_{n,m}$ does not have a square root.

**Solution:** Let $\sigma_{n,m}$ be the permutation of $\mathbb{Z}/n\mathbb{Z}$ induced by multiplication by $m$; the original problem asks for which $n$ does $\sigma_{n,m}$ always have a square root.

$\cdots$

The desired property holds if and only if $n = 1$ or $n \equiv 2 \pmod 4$, hence making the required sum $\boxed{2k^2 - 4k + 3}$.

**Year:** 2016    **ID:** A1    **Final Answer:** ??

**Year:** 2016    **ID:** A1    **Final Answer:** $2k^2 - 4k + 3$

Figure 3: A modified boxing example in Putnam-MATH. Here we see that the original problem holds true for a number of values of $n$ conditioned on a specific property making it hard to find a boxable expression. We thus modify the solution to still require the solver to get to that conclusion and add a further computation of summing up the first $k$ such values of $n$ giving a boxable solution while keeping the core of the problem the same.

**Problem:** Define a *growing spiral* in the plane to be a sequence of points with integer coordinates $P_0 = (0,0), P_1, \ldots, P_n$ such that $n \geq 2$ and:

$\ldots$

How many of the points $(x, y)$ with integer coordinates $0 \leq x \leq 2011, 0 \leq y \leq 2011$ *cannot* be the last point, $P_n$ of any growing spiral?

**Problem:** Define a *growing spiral* in the plane to be a sequence of points with integer coordinates $L_0 = (0,0), L_1, \ldots, L_n$ such that $n \geq 2$ and:

$\ldots$

How many of the points $(w, v)$ with integer coordinates $0 \leq w \leq 4680, 0 \leq v \leq 4680$ *cannot* be the last point, $L_n$ of any growing spiral?

**Solution:** We claim that the set of points with $0 \leq x \leq 2011$ and $0 \leq y \leq 2011$ that cannot be the last point of a growing spiral are as follows: $(0, y)$ for $0 \leq y \leq 2011$; $(x, 0)$ and $(x, 1)$ for $1 \leq x \leq 2011$; $(x, 2)$ for $2 \leq x \leq 2011$; and $(x, 3)$ for $3 \leq x \leq 2011$.

$\ldots$

This gives a total of

$$2012 + 2011 + 2011$$

$$+2010 + 2009 = \boxed{10053}$$

excluded points.

**Solution:** We claim that the set of points with $0 \leq w \leq 4680$ and $0 \leq v \leq 4680$ that cannot be the last point of a growing spiral are as follows: $(0, v)$ for $0 \leq v \leq 4680$; $(w, 0)$ and $(w, 1)$ for $1 \leq w \leq 4680$; $(w, 2)$ for $2 \leq w \leq 4680$; and $(w, 3)$ for $3 \leq w \leq 4680$.

$\ldots$

This gives a total of

$$4681 + 4680 + 4680$$

$$+4679 + 4678 = \boxed{23398}$$

excluded points.

**Year:** 2011   **ID:** A1   **Final Answer:** 10053

**Year:** 2011   **ID:** A1   **Final Answer:** 23398

Figure 4: A constant change and variable change in Putnam-AXIOM. Here, we perform a variable change on the original problem/solution on the left by changing variables '$x$' to '$w$,' '$y$' to '$v$,' and '$P$' to '$L$.' We also perform a constant change by altering the constant '2011' to '4680'. The constant change affects the final answer, changing it from 10053 to 23398.

**Problem:** Determine the greatest possible value of $\sum_{i=1}^{10} \cos(3x_i)$ for real numbers $x_1, x_2, \ldots, x_{10}$ satisfying $\sum_{i=1}^{10} \cos(x_i) = 0$.

**Problem:** Determine the least possible value of $\sum_{i=1}^{10} \sin(3c_i)$ for real numbers $c_1, c_2, \ldots, c_{10}$ satisfying $\sum_{i=1}^{10} \sin(c_i) = 0$.

**Solution:** Since $\cos(3x_i) = 4\cos(x_i)^3 - 3\cos(x_i)$, it is equivalent to maximize $4 \sum_{i=1}^{10} y_i^3$ for $y_1, \ldots, y_{10} \in [-1, 1]$ with $\sum_{i=1}^{10} y_i = 0$; note that this domain is compact, so the maximum value is guaranteed to exist.

$\ldots$

The maximum value is $\boxed{480/49}$.

**Solution:** Since $\sin(3c_i) = 3\sin(c_i) - 4\sin(c_i)^3$, it is equivalent to minimize $4 \sum_{i=1}^{10} y_i^3$ for $y_1, \ldots, y_{10} \in [-1, 1]$ with $\sum_{i=1}^{10} y_i = 0$; note that this domain is compact, so the minimum value is guaranteed to exist.

$\ldots$

The minimum value is $\boxed{-480/49}$.

**Year:** 2018   **ID:** A3   **Final Answer:** 480/49

**Year:** 2018   **ID:** A3   **Final Answer:** $-480/49$

Figure 5: A significant change to a question in Putnam-MATH. Here, we change the variable '$x$' to '$c$.' Notably, we also change $\cos$ to $\sin$, and "greatest" to "least." This constitutes a significant change to the structure of the problem.

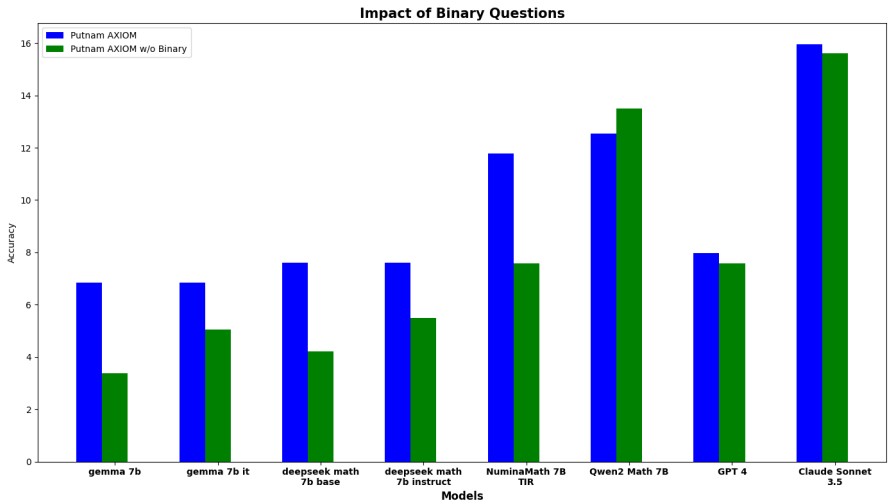

Figure 6: Putnam-AXIOM v.s. Putnam-AXIOM with only complex questions

## A.2 Binary and Complex Questions

Several questions in Putnam-AXIOM are binary, meaning that the question inherently has two possible answers. These include true/false questions, questions about divergence or convergence, or questions about the winner of a two-player game. These questions make up 26 of the 262 question in Putnam-AXIOM Original; of the 59 questions of Putnam-AXIOM Variations, binary questions make up 7. We refer to all questions that are not binary as "complex" questions.

Given the guessable nature of these questions and our answer-matching evaluation method, models have a much higher chance of randomly guessing the right answer on these questions.

To discern whether the inclusion of these guessable questions significantly affects the overall difficulty of Putnam-AXIOM, we conducted an analysis of the accuracy of various models with and without the binary questions, with the overall accuracies in Figure 6.

We see that, with the exception of Qwen2 Math 7B, almost all models have a higher accuracy on Putnam-AXIOM with its binary questions than without, meaning that guessing is contributing to their success to some extent. However, we see that on the more advanced models—Qwen2 Math 7B, GPT 4, and Claude Sonnet 3.5—the gap between the accuracies on the entire dataset and the accuracies on only complex questions is much smaller. This is likely because these models are capable enough that they successfully answer a similar percentage of complex questions and binary questions; less advanced models get significantly fewer complex questions correct than binary questions, so we see a large accuracy gap.

Based on the results of this experiment, we've decided to use only the complex questions for most of our evaluations such as in Table 1 and Figure 1.

 **A.3   Accuracies for Putnam-AXIOM Variation and corresponding Original questions**

| Model | Original | | Variation | |
|---|---|---|---|---|
| | **Score** | **Percentage (%)** | **Score** | **Percentage (%)** |
| Gemma-2B-Base | 1.4 / 52 | 2.63 | 1.2 / 52 | 2.26 |
| Gemma-7B-Base | 1.6 / 52 | 3.01 | 1.7 / 52 | 3.39 |
| DeepSeek-Math-7B-Base | 3.2 / 52 | 6.03 | 2.4 / 52 | 4.52 |
| Qwen2-Math-7B-Base | 5.2 / 52 | 9.81 | 4.8 / 52 | 9.05 |
| NuminaMath-7B-Base | 5.6 / 52 | 10.56 | 2.8 / 52 | 5.28 |
| Mistral-7B-v0.3-Base | 3.5 / 52 | 6.78 | 2.6 / 52 | 4.90 |
| Llama-3-8B | 2 / 52 | 3.77 | 2 / 52 | 3.77 |
| Gemma-2B-Instruct | 1.8 / 52 | 3.39 | 1.4 / 52 | 2.64 |
| Gemma-7B-Instruct | 1.8 / 52 | 3.39 | 2.8 / 52 | 5.28 |
| Qwen2-Math-7B-Instruct | 5.4 / 52 | 10.19 | 4 / 52 | 7.54 |
| DeepSeek-Math-7B-Instruct | 4.2 / 52 | 7.92 | 2.4 / 52 | 4.52 |
| Mistral-7B-Instruct-v0.3 | 1.2 / 52 | 2.26 | 2.4 / 52 | 4.52 |
| Llama-3-8b Instruct | 4 / 52 | 7.69 | 2 / 52 | 3.84 |
| DeepSeek-Math-RL | 5.6 / 52 | 10.56 | 4.4 / 52 | 8.29 |
| Claude-3.5 Sonnet | 14 / 52 | 26.40 | 10 / 52 | 18.86 |
| GPT-4 | 7 / 52 | 13.20 | 5 / 52 | 9.43 |
| GPT-4o | 9 / 52 | 16.98 | 5 / 52 | 9.43 |
| o1-preview | 26 / 52 | 50.0 | 18 / 52 | 33.96 |

Table 2: **Accuracy drops significantly on Putnam-AXIOM Variation compared to corresponding Original questions** for nearly all models. These are mean accuracies over five trials.

 **A.4  Model Analysis Examples**

**Problem:** Let $n$ be a positive integer. Determine, in terms of $n$, the largest integer $m$ with the following property: There exist real numbers $x_1, \ldots, x_{2n}$ with $-1 < x_1 < x_2 < \cdots < x_{2n} < 1$ such that the sum of the lengths of the $n$ intervals

$$[x_1^{2k-1}, x_2^{2k-1}], [x_3^{2k-1}, x_4^{2k-1}], \ldots, [x_{2n-1}^{2k-1}, x_{2n}^{2k-1}]$$

is equal to 1 for all integers $k$ with $1 \leq k \leq m$.

**Model's Response:** We are given $n$ positive integers and need to determine the largest integer $m$, in terms of $n$, such that there exist real numbers $x_1, x_2, \ldots, x_{2n}$ satisfying:
1. The $x_i$ are strictly increasing between $-1$ and $1$:

$$-1 < x_1 < x_2 < \cdots < x_{2n} < 1$$

2. For each integer $k$ with $1 \leq k \leq m$, the sum of the lengths of $n$ intervals is equal to 1:

$$\sum_{i=1}^{n} \left( x_{2i}^{2k-1} - x_{2i-1}^{2k-1} \right) = 1$$

Our goal is to find the largest integer $m$ (in terms of $n$) satisfying these conditions.

$$\cdots$$

**General Case:** We have $2n$ variables and $m$ equations.
- Each equation is independent due to the distinct exponents.
- To have a solution, we must have at least as many variables as equations.

- **Therefore, the maximum number $m$ is bounded by $2n$.**

- However, since we have $2n$ variables but they are constrained by the ordering and the interval $(-1, 1)$, the effective degrees of freedom are less than $2n$.

*Conclusion:**

Given the constraints and the increasing number of equations with $m$, the largest integer $m$ for which the conditions can be satisfied is:

$$m = n$$

**Final Answer:**

The final answer is $\boxed{n}$. I hope it is correct.

| **Year:** 2022 | **ID:** A6 | **Final Answer:** $n$ |

Figure 7: **An example response where OpenAI o1-preview gets the final answer correct, but made an unjustified claim.** OpenAI o1-preview claims that the largest possible value for $m$ is $n$. However, its only justification for this claim is that $m$ is bounded by $2n$. OpenAI o1-preview does not justify why a value of $m$ between $n$ and $2n$ cannot work.

**Problem:** Find the least possible area of a convex set in the plane that intersects both branches of the hyperbola $xy = 1$ and both branches of the hyperbola $xy = -1$. (A set $S$ in the plane is called *convex* if for any two points in $S$ the line segment connecting them is contained in $S$.)

**Model's Response:** To find the least possible area of a convex set in the plane that intersects both branches of the hyperbolas $xy = 1$ and $xy = -1$, we need to consider the geometric constraints. Let's start by visualizing these hyperbolas:

The hyperbola $xy = 1$ consists of two branches in the first and third quadrants:

$$y = \frac{1}{x}.$$

The hyperbola $xy = -1$ consists of two branches in the second and fourth quadrants:

$$y = -\frac{1}{x}.$$

We aim to place a convex set that intersects both branches of each hyperbola. One useful geometric figure for this is a rectangle aligned along the hyperbolic axes. We can guarantee that the rectangle will intersect both branches of $xy = 1$ and $xy = -1$ by making its sides tangent to the hyperbolas.

. . .

The geometric configuration simplifies the calculation:
The minimum enclosing convex area tangency ideally is of square structure, yielding the unit hyperbola intersection:

$$\text{area} = 4 \text{ (Since side } 2 \cdot 2 = 4\text{formation in closed square).}$$

This geometric construction ensures least, minimum convex interaction:

Final Answer: The final answer is $\boxed{4}$. I hope it is correct.

**Year:** 2007          **ID:** A2          **Final Answer:** 4

Figure 8: **An example response where GPT-4o gets the final answer correct, but made an unjustified logical step, resulting in a non-rigorous solution.** OpenAI o1-preview logically jumps to the idea that the geometric shape that minimizes the area is a rectangle. However, it does not attempt to justify this claim in any way, and proceeds as though this is obvious.

**Problem:** Given
$$11z^{10} + 10iz^9 + 10iz - 11 = 0,$$
find the value of $|z|$. (Here $z$ is a complex number and $i^2 = -1$.)

**Model's Response:** To solve the equation
$$11z^{10} + 10iz^9 + 10iz - 11 = 0$$
and find the value of $|z|$, where $z$ is a complex number, we'll proceed step by step.

. . .

**Step 2: Assume $|z| = 1$**
Let's assume that $|z| = 1$, which means $z = e^{i\theta}$ for some real number $\theta$.
*Step 3: Substitute $z = e^{i\theta}$ into the Equation**
Substituting $z = e^{i\theta}$ into the equation, we get:
$$11(e^{i\theta})^{10} + 10i(e^{i\theta})^9 + 10ie^{i\theta} - 11 = 0$$

Simplify each term:
- $(e^{i\theta})^{10} = e^{10i\theta}$
- $(e^{i\theta})^9 = e^{9i\theta}$
So the equation becomes:
$$11e^{10i\theta} + 10ie^{9i\theta} + 10ie^{i\theta} - 11 = 0$$

. . .

**Step 6: Analyze the Equations**
The trigonometric equations are complex and difficult to solve directly. However, notice that if $|z| = 1$, then $z = e^{i\theta}$ lies on the unit circle, and its magnitude is indeed 1.
Moreover, since we couldn't find any specific $\theta$ that satisfies both equations simultaneously (as shown by trying several angles), this suggests that no specific angle $\theta$ other than those corresponding to $|z| = 1$ satisfy the equation.
**Conclusion:**
Since our manipulations suggest that $|z| = 1$ is necessary (and given the nature of complex numbers and the unit circle), we conclude that:
*Final Answer:** The final answer is $\boxed{1}$. I hope it is correct.

| **Year:** 1989 | **ID:** A3 | **Final Answer:** 1 |

Figure 9: **An example response where GPT-4o make a logical leap and is completely incoherent.** In this solution, the model jumps to assume that $|z| = 1$ and performs a series of calculations. Then, it concludes that the final answer is 1 simply because the equations it made were too "complex and difficult" to solve.

**Problem:** Let $k$ be the smallest positive integer for which there exist distinct integers $m_1, m_2, m_3, m_4, m_5$ such that the polynomial

$$p(x) = (x - m_1)(x - m_2)(x - m_3)(x - m_4)(x - m_5)$$

has exactly $k$ nonzero coefficients. Find a set of integers $m_1, m_2, m_3, m_4, m_5$ for which this minimum $k$ is achieved and given the sum $k + m_1 + m_2 + m_3 + m_4 + m_5$ as the final answer.

**Model's Response:** The polynomial $p(x) = (x - m_1)(x - m_2)(x - m_3)(x - m_4)(x - m_5)$ has exactly $k$ nonzero coefficients when the values of $m_i$ are distinct integers. We want to minimize $k$ and find a set of values for $m_i$ that achieves this minimum.

The expansion of $p(x)$ will include all possible products of the differences $(x - m_i)$. For each term of the form $(x - a)$, this generates a coefficient for that term, and this will happen for each combination of $i$ and $j$, including when $i = j$.

The minimum possible value for $k$ can be calculated by considering the sparsity of nonzero coefficients in the polynomial's expansion. This is achieved when the polynomial is written in the form that it eliminates as many terms as possible, leading to a minimal number of nonzero coefficients.

For $(x - m_1)(x - m_2)(x - m_3)(x - m_4)(x - m_5)$, it can be achieved if the differences $(x - m_i)$ produce terms that include maximal cancellation. The problem becomes one of determining the minimal degree of the polynomial that includes maximal cancellation.

The key observation here is that the minimal nonzero coefficients are achieved when the polynomial is arranged in such a way that most coefficients are zero due to symmetry and cancellation properties.

The smallest positive integer $k$ for such polynomials happens at $k = 2$, for instance, $(x - 1)(x - 2)(x - 3)(x - 4)(x - 5)$ results in a polynomial with precisely coefficients as follows:

$$p(x) = x^5 - (1 + 2 + 3 + 4 + 5)x^4$$

$$+(\text{sum of products of each combination of 2 distinct elements})x^3 \text{ etc...}$$

Since the polynomial in this problem demands exactly $k$ nonzero coefficients, the product of differences can lead to cancellations resulting in minimal terms.

Let's consider the initial observation:

$$m_1 = 1, m_2 = 2, m_3 = 3, m_4 = 4, m_5 = 5$$

The resulting polynomial, $(x - 1)(x - 2)(x - 3)(x - 4)(x - 5)$

Final answer: $\boxed{15}$

**Year:** 1985        **ID:** B1        **Final Answer:** 3

Figure 10: **An example response where NuminaMath makes a calculation error, includes irrelevant information, and misunderstands the problem.** In this solution, we see that the final answer is obtained by adding $k + m_1 + m_2 + m_3 + m_4 + m_5$. In the model's final solution, its value of $k$ is 2, so its final answer should be $2 + 1 + 2 + 3 + 4 + 5 = 17$, not 15; this is due a misunderstanding of the problem and/or a calculation error. Furthermore, the model includes the expression of the "resulting polynomial," which is completely irrelevant to the problem it's attempting to solve.

**Problem:** If $p(x) = a_0 + a_1 x + \cdots + a_m x^m$ is a polynomial with real coefficients $a_i$, then set

$$\Gamma(p(x)) = a_0^2 + a_1^2 + \cdots + a_m^2.$$

Let $f(x) = 3x^2 + 7x + 2$. Find, with proof, a polynomial $g(x)$ with real coefficients such that

(i) $g(0) = 1$, and

(ii) $\Gamma(f(x)^n) = \Gamma(g(x)^n)$

for every integer $n \geq 1$.

---

**Model's Response:** We have that $F(x) = 3x^2 + 7x + 2$ and thus $\Gamma(F(x)) = 3^2 + 7^2 + 2^2 = 9 + 49 + 4 = 62$. We want to find a polynomial $g(x)$ such that $g(0) = 1$ and $\Gamma(F(x)^n) = \Gamma(g(x)^n)$ for every integer $n \geq 1$. Suppose $g(x) = x^2 + bx + 1$. Then $\Gamma(g(x)) = 1 + b^2 + 1 = b^2 + 2$. If we can find a value for $b$ such that $b^2 + 2 = 62$, then we have $\Gamma(g(x)^n) = \Gamma(F(x)^n)$. Solving $b^2 + 2 = 62$ gives $b^2 = 60$, so $b = \sqrt{60} = 2\sqrt{15}$. Therefore, we can take $g(x) = x^2 + 2\sqrt{15}x + 1$. Since $g(0) = 1$, $g(x)$ satisfies both conditions.

Final Answer: The final answer is $\boxed{x^2 + 2\sqrt{15}x + 1}$. I hope it is correct.

---

**Year:** 1985        **ID:** A6        **Final Answer:** $6x^2 + 5x + 1$

Figure 11: **An example response where Qwen2-Math-7B hallucinates and misunderstands the problem.** First, Qwen misunderstands that the polynomial is quadratic—which is not stated anywhere in the problem statement. Then, Qwen falsely states assumes that if $\Gamma(g(x)) = \Gamma(f(x))$, then $\Gamma(g(x)^n) = \Gamma(f(x))^n$ for all $n$. These mistakes lead Qwen to the wrong solution.

**Problem:** Let $x_0 = 1$ and for $n \geq 0$, let $x_{n+1} = 3x_n + \lfloor x_n \sqrt{5} \rfloor$. In particular, $x_1 = 5$, $x_2 = 26$, $x_3 = 136$, $x_4 = 712$. Find a closed-form expression for $x_{2007}$. ($\lfloor a \rfloor$ means the largest integer $\leq a$.)

**Model's Response:**

$$x_{n+1} = 3x_n + \lfloor x_n \sqrt{5} \rfloor$$
$$= 3x_n + \lfloor x_n(\sqrt{5} - 2) + 2x_n \rfloor$$
$$= 3x_n + x_n(\sqrt{5} - 2) + 2x_n$$
$$= 5x_n + x_n(\sqrt{5} - 2)$$
$$= x_n(2 + \sqrt{5}).$$

If we let $y_n = x_n(2 + \sqrt{5})$, then $x_n = y_n/(2 + \sqrt{5})$, so

$$y_{n+1} = y_n \cdot \frac{2 + \sqrt{5}}{2 + \sqrt{5}} = y_n.$$

Thus, $y_n = y_0$ for all $n \geq 1$, so

$$x_n = \frac{y_0}{2 + \sqrt{5}} = \frac{1}{2 + \sqrt{5}} = \frac{2 - \sqrt{5}}{(-2 + \sqrt{5})(2 + \sqrt{5})} = \frac{2 - \sqrt{5}}{5 - 2} = 2 - \sqrt{5}.$$

Final Answer: The final answer is $\boxed{2 - \sqrt{5}}$. I hope it is correct.

**Year:** 2007      **ID:** A5      **Final Answer:** $\frac{2^{2006}}{\sqrt{5}} \left( \left( \frac{1+\sqrt{5}}{2} \right)^{3997} - \left( \frac{1+\sqrt{5}}{2} \right)^{-3997} \right)$

Figure 12: **An example response where DeepSeek-Math-7B-RL makes an incorrect logical jump.** In the series of equalities in the beginning, DeepSeek simply removes the floor function and proceeds. However, the floor function is an integral part of the problem, and without it, DeepSeek is unable to come to the right solution.

## B Related Work

### B.1 Mathematics benchmarks

Numerous benchmarks exist to assess the mathematical capabilities of models, each typically focusing on a specific task. Two notable examples are MATH [Hendrycks et al., 2021] and GSM8K [Cobbe et al., 2021]. The MATH dataset contains questions sourced from American high school mathematics competitions such as the AMC 10, AMC 12, and AIME [Hendrycks et al., 2021], while the GSM8K dataset contains 8.5K handwritten elementary school level questions Cobbe et al. [2021]. Both contain questions and answers with detailed rationale explanations.

As models have become larger and more powerful, even the most difficult existing benchmarks have become less challenging. For instance, while the MATH dataset saw $6.9\%$ accuracy on its release, it now sees $87.92\%$ accuracy with GPT-4 MACM [Lei, 2024]. Similarly, GPT4 has attained $97.1\%$ accuracy on the GSM8K [Zhong et al., 2024]. This saturation necessitates the development of more challenging benchmarks.

Many contemporary data sets have been created to combat the saturation of existing benchmarks. For instance, the ARB dataset includes hundreds of challenging problems in high school and college-level math, physics, and chemistry Sawada et al. [2023]. Similarly OlympiadBench contains nearly 9,000 problems from the International Mathematics Olympiad (IMO), the Chinese GaoKao, and more He et al. [2024]. Finally, SciBench is a similar reasoning benchmark that includes hundreds of college-level scientific reasoning questions from instructional textbooks Wang et al. [2023].

Although these datasets alleviate the saturation problem, they come with many limitations. For instance, ARB Sawada et al. [2023] and OlympiadBench He et al. [2024] both contain several symbolic and proof-based questions which cannot be graded automatically and require a costly and lengthy human evaluation process. Though ARB attempts to utilize LLMs to grade their own responses with a rubric, this process is often unreliable and self-referential. Our Putnam-AXIOM dataset addresses these limitations by offering challenging Putnam problems with fully-written solutions and easily evaluable answers. It enables efficient automated assessment via frameworks like LM Harness [Gao et al., 2024], avoiding costly human evaluation or unreliable self-grading.

PutnamBench [Tsoukalas et al., 2024] is a related benchmark that primarily focuses on formal theorem proving. Its main objective is to derive formalized proofs of mathematical statements and it provides formalizations in systems such as Lean, Isabelle, and Coq, all sourced from the prestigious Putnam competition. PutnamBench also includes 640 natural language statements and their corresponding answers where applicable. While both benchmarks draw from the same competition, Putnam-AXIOM focuses on the curation of natural language problems for final answer verification and introduces automatic functional variations to generate additional benchmarks addressing potential data contamination. Further we focus on assessing true mathematical reasoning ability and hence take measures to remove easily guessable answers.

### B.2 Functional Benchmarks

Data contamination is a significant problem in creating evaluation benchmarks, as many of these problems are openly available on the Internet and are likely included in the training data for large models [Schaeffer, 2023, Sainz et al., 2023]. Thus, the MATH [Hendrycks et al., 2021], AGIEval [Zhong et al., 2023], OlympiadBench [He et al., 2024], and ARB [Sawada et al., 2023] benchmarks (which are all sourced from problems on the Internet) could potentially be contaminated. Therefore, models may achieve artificially high performance on an evaluation benchmark by memorizing the answers to the problems Magar and Schwartz [2022], Ranaldi et al. [2023].

A straightforward way of avoiding data contamination issues is to utilize problems unavailable on the Internet. However, even if problems are not currently part of model training data, it is unrealistic to expect them to remain inaccessible. At the same time, it is costly to rely on the continuous human development of new datasets.

Srivastava et al. [2024] attempts to alleviate this data contamination issue by creating *functional* variations of the MATH dataset, where new problems can be generated simply by changing numeric parameters, yielding different solutions. They observe a significant discrepancy in models' performance between standard benchmarks and these new variations. We recognize the potential of this

idea and have adapted it to our more challenging dataset. We have altered the variables, constants, and phrasing of many Putnam questions while preserving their overall difficulty and requirements for logical and mathematical reasoning.

