# OpenReview forum: "Putnam-AXIOM: A Functional and Static Benchmark for Measuring Higher Level Mathematical Reasoning"
_NeurIPS.cc/2024/Workshop/MATH-AI — MATH-AI 24_

### Official Review · Reviewer_MsMi · 2024-10-06
**Review of "Putnam-AXIOM"**

**Rating:** 7
**Confidence:** 4

**Review:**

**Summary**

This paper presents a new benchmark, Advanced Examination of Intelligence in Operational Mathematics (Putnam-AXIOMO).
This benchmark consists of 236 problems which can be automatically evaluated as the answers must be provided inside of a
"\boxed{}" command.
Alongside releasing the anonymous code, the authors also deal with dataset contamination, as these problems have been publicly available, by mutating parts of 53 problems so that they cannot have been memorised by any model.

**Strengths**
- New difficult reasoning benchmark, even very strong models struggle to perform well on this benchmark.
- Deals with training set contamination. One of the largest problems for current benchmarks is train set contamination, it is nice to see it concisely dealt with and the decontamination process yields accuracy decreases for almost all models.

**Weaknesses**
- Requires the models to use "\boxed{}" for their final answer limiting the expressivity of the benchmark.
- Minor point: please use bigger font sizes in the plots.

**Questions**
- How many of the problems were marked incorrect because the model did not use the \boxed command correctly? I would like to understand more the difference between the models ability to follow instructions (use \boxed like it is instructed to) and its ability to reason.
- Is there an algorithmic way to keep editing these problems so that there is always a dataset we can be sure no model has memorised?

---

### Official Review · Reviewer_Nbvs · 2024-10-06
**Good benchmark contribution**

**Rating:** 7
**Confidence:** 3

**Review:**

This work proposes a mathematical benchmark based on challenging Putnam questions to evaluate the mathematical reasoning abilities of large language models (LLMs). The experimental result show that current strong LLMs perform poorly on this benchmark, and is negatively affected by the reasoning irrelevant variations like variable change or constant change, indicating potential contamination of the data.

In general, this benchmark would be a good contribution for more challenging problem set to evaluate the mathematical problem solving abilities (with rich examples and explainations regarding potential legal concerns). The problem modifications would be a good strategy to reduce problem loss in terms of the current box-based evaluation scheme (maybe it could be more scalable if detailed explained). Besides, there are some existing work on revealing the potential performance gap that might reflect data contamination like [1], would findings in those works echo or strengthens the takeaways here? Also, how would this dataset being a valuable adding-on to existing benchmarks that are based on challenging examinations like [2]?

[1] Xu et al. Benchmarking Benchmark Leakage in Large Language Models. https://arxiv.org/pdf/2404.18824v1

[2] Huang et al. OlympicArena: Benchmarking Multi-discipline Cognitive Reasoning for Superintelligent AI. NeurIPS Dataset and Benchmark 2024. https://arxiv.org/pdf/2406.12753

---

### Official Review · Reviewer_krr4 · 2024-10-07

**Rating:** 5
**Confidence:** 4

**Review:**

The motivation of the paper is interesting. However, the dataset only consists of 236 examples, which is too small to serve as a convincing benchmark. Also, most evaluated models achieve very low accuracy, which questions the effectiveness and robustness - the difficulty of the problems should be more hierarchical.

---

### Official Review · Reviewer_9XA2 · 2024-10-08
**Benchmarks suffers from low power and contamination**

**Rating:** 5
**Confidence:** 4

**Review:**

The paper introduces a benchmark based on 236 Putnam problems between 1985 and 2023. The authors meticulously filtered problems that are correctly gradable ensuring minimizing noise from
- problem formatting (latex equations, images)
- boxed answer construction
- answer equality checking

Next, the authors identify contamination as a problem in benchmark evaluation considering the time horizon of the problems collected. They proposed functional problem transformations as a potential approach to fix this. However, I believe that this level of contamination prevention might be insufficient as the problem can also memorize problem-specific logic beyond simply memorizing the answer. Additionally, functional transformations are only applied on 53 problems which reduces the power of evaluations and would like result in high error bars during evaluation.

---

### Decision · Program_Chairs · 2024-10-09

Accept